# Detecting and representing predictable structure during auditory scene analysis

**Ediz Sohoglu\*, Maria Chait\***

UCL Ear Institute, University College London, London, United Kingdom

**Abstract** We use psychophysics and MEG to test how sensitivity to input statistics facilitates auditory-scene-analysis (ASA). Human subjects listened to 'scenes' comprised of concurrent tone-pip streams (sources). On occasional trials a new source appeared partway. Listeners were more accurate and quicker to detect source appearance in scenes comprised of temporally-regular (REG), rather than random (RAND), sources. MEG in passive listeners and those actively detecting appearance events revealed increased sustained activity in auditory and parietal cortex in REG relative to RAND scenes, emerging ~400 ms of scene-onset. Over and above this, appearance in REG scenes was associated with increased responses relative to RAND scenes. The effect of temporal structure on appearance-evoked responses was delayed when listeners were focused on the scenes relative to when listening passively, consistent with the notion that attention reduces 'surprise'. Overall, the results implicate a mechanism that tracks predictability of multiple concurrent sources to facilitate active and passive ASA.

\*For correspondence: e.sohoglu@
gmail.com (ES); m.chait@ucl.ac.uk
(MC)

**Competing interests:** The
authors declare that no
competing interests exist.

**Reviewing editor:** Andrew J
King, University of Oxford,
United Kingdom

## Introduction

Natural scenes are highly structured, containing statistical regularities in both space and time and over multiple scales (*Julesz, 1981*; *Portilla and Simoncelli, 2000*; *Geisler, 2008*; *McDermott et al., 2013*; *Theunissen and Elie, 2014*). A growing body of work suggests that the human brain is sensitive to this statistical structure (*Rao and Ballard, 1999*; *Näätänen et al., 2001*; *Bar, 2004*; *Oliva and Torralba, 2007*; *Costa-Faidella et al., 2011*; *Garrido et al., 2013*; *Okazawa et al., 2015*; *Barascud et al., 2016*) and uses it for efficient scene analysis (*Winkler et al., 2009*; *Andreou et al., 2011*; *Bendixen, 2014*). Uncovering the process by which this occurs, and how sensory predictability interacts with attention, is a key challenge in sensory neuroscience across modalities (*Winkler et al., 2009*; *Summerfield and de Lange, 2014*; *Summerfield and Egner, 2016*).

The current state of understanding is limited by at least two factors: (1) most studies of sensory predictability and its effects on behavior have used slow presentation rates thus enabling conscious reflection of stimulus expectancy. As a consequence, relatively little is known about the neural underpinning of predictability processing on the rapid time scales relevant to perception of natural objects. (2) In most cases, predictability has been studied when participants attend to a single object (*Murray et al., 2002*; *Arnal et al., 2011*; *Kok et al., 2012*; *Chennu et al., 2013*; *Bendixen, 2014*) – a far cry from the complex scenes in which we normally operate. We therefore do not understand whether/how statistical structure is extracted from complex, crowded scenes. The present work addresses both of these issues in the context of an auditory scene.

To understand how statistical structure facilitates perceptual analysis of acoustic scenes, we use an ecologically relevant paradigm (change detection) that captures the challenges of natural listening in crowded environments (*Cervantes Constantino et al., 2012*; *Sohoglu and Chait, 2016*). In this paradigm, listeners are presented with multiple concurrent acoustic sources and on occasional trials, a new source appears partway into the ongoing scene (see *Figure 1A*). By varying the temporal patterning of scene sources, we can create conditions in which the scenes are characterized by

**eLife digest** Everyday environments like a busy street bombard our ears with information. Yet most of the time, the human brain quickly and effortlessly makes sense of this information in a process known as auditory scene analysis. According to one popular theory, the brain is particularly sensitive to regularly repeating features in sensory signals, and uses those regularities to guide scene analysis. Indeed, many biological sounds contain such regularities, like the pitter-patter of footsteps or the fluttering of bird wings.

In most previous studies that investigated whether regularity guides auditory scene analysis in humans, listeners attended to one sound stream that repeated slowly. Thus, it was unclear how regularity might benefit scene analysis in more realistic settings that feature many sounds that quickly change over time.

Sohoglu and Chait presented listeners with cluttered, artificial auditory scenes comprised of several sources of sound. If the scenes contained regularly repeating sound sources, the listeners were better able to detect new sounds that appeared partway through the scenes. This shows that auditory scene analysis benefits from sound regularity.

To understand the neurobiological basis of this effect, Sohoglu and Chait also recorded the brain activity of the listeners using a non-invasive technique called magnetoencephalography. This activity increased when the sound scenes featured regularly repeating sounds. It therefore appears that the brain prioritized the repeating sounds, and this improved the ability of the listeners to detect new sound sources.

When the listeners actively focused on listening to the regular sounds, their brain response to new sounds occurred later than seen in volunteers who were not actively listening to the scene. This was unexpected as delayed brain responses are not usually associated with active focusing. However, this effect can be explained if active focusing increases the expectation of new sounds appearing, because previous research has shown that expectation reduces brain responses.

The experiments performed by Sohoglu and Chait used a relatively simple form of sound regularity (tone pips repeating at equal time intervals). Future work will investigate more complex forms of regularity to understand the kinds of sensory patterns to which the brain is sensitive.

statistically regular or random structure and measure the effect of this manipulation on listeners' ability to detect the appearance of new sources within the unfolding soundscape.

The behavioral response pattern reveals that perceptual analysis of such scenes is enhanced by the presence of regular statistical structure, as assessed by listeners' ability to detect source appearance. One possible explanation for this effect is that neural responses to regularly repeating scene components adapt (decrease over time) more than to random components. Indeed, perceptual influences of statistical structure have often been attributed to neural adaptation (e.g. 'stimulus specific adaptation'; *May et al., 1999*; *Jääskeläinen et al., 2004*; *Haenschel et al., 2005*; *Costa-Faidella et al., 2011*; ; *Khouri and Nelken, 2015*). Accordingly, the relative change in neural response to a new spectral component (that is, the appearing source) will be larger and thus more detectable in regular versus random scenes (*Summerfield et al., 1987*; *Hartmann and Goupell, 2006*; *Erviti et al., 2011*). By this account, statistical structure does not modulate the magnitude of neural response to a new event per se. Rather, improved detection is attributed exclusively to decreased neural responses occurring before the appearance of the new source. Indeed, in a mismatch negativity paradigm, *Costa-Faidella (2011)* demonstrated that neural responses to repeating ('standard') tones adapt more in temporally regular than random sequences without accompanying changes in response to new ('deviant') tones (see also *Schwartze et al., 2011*, *2013*; *Tavano et al., 2014*).

However, other work has shown that statistically regular patterns can be associated with increased neural responses (*Haenschel et al., 2005*; *Kok et al., 2012*; *Chennu et al., 2013*; *Hsu et al., 2014*; *Kouider et al., 2015*; *Barascud et al., 2016*). These effects have been interpreted to reflect a mechanism that tracks the level of predictability or 'precision' of the sensory input, a measure inversely related to the uncertainty or entropy of a variable. This mechanism is hypothesized

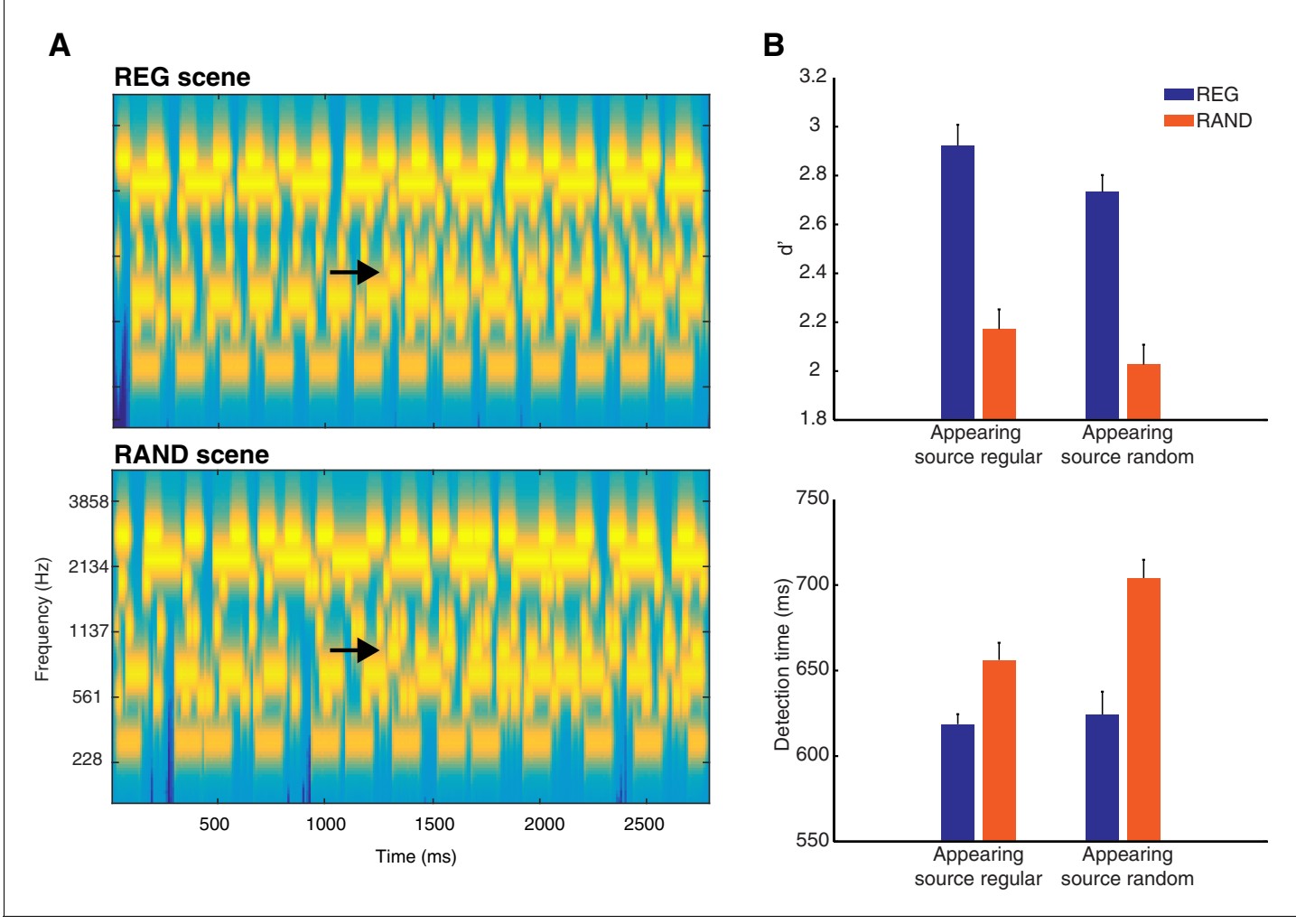

**Figure 1.** Stimuli and behavior. (**A**) Examples of REG and RAND scenes. The plots represent 'auditory' spectrograms, equally spaced on a scale of ERB-rate (**Moore and Glasberg, 1983**). Channels are smoothed to obtain a temporal resolution similar to the Equivalent Rectangular Duration (**Plack and Moore, 1990**). Black arrows indicate appearing sources. In these examples, the appearing source is temporally regular. The stimulus set also included scenes in which the appearing source was temporally random (see Materials and methods). (**B**) Behavioral results (d' and detection time) as a function of scene temporal structure (REG versus RAND). These are shown for each type of scene change (when the appearing source was temporally regular or when random). Error bars represent within-subject standard error of the mean (SEM; **Loftus and Masson, 1994**).

to enable the up-regulation of processing for information that is reliable and likely to indicate genuine events in the environment (**Feldman and Friston, 2010**; **Zhao et al., 2013**; **Auksztulewicz and Friston, 2015**; **Barascud et al., 2016**). Importantly, the up-regulation of neural processing in these accounts is hypothesized to lead to increased neural responses for regular scenes before source appearance, as well as an increased error ('surprise') response evoked by the new source.

In the current study we adjudicate between adaptation and precision accounts using magnetoencephalography (MEG) recordings of brain activity. Given the ongoing debate about how the neural influence of statistical regularity might depend on attention (**Jones and Boltz, 1989**; **Näätänen et al., 2001**; **Summerfield and Egner, 2009**; **Winkler et al., 2009**; **Feldman and Friston, 2010**; **Kok et al., 2012**; **Bendixen, 2014**; **Schröger et al., 2015**), we do this in the context of passive listening (listeners engaged in an unrelated visual task) as well as active listening (listeners actively detecting source appearance). Our results provide evidence in support of precision accounts: we show that brain responses to ongoing acoustic scenes, and to new sources appearing within those scenes, increase in the presence of regular statistical structure. Strikingly, the effect of regularity on appearance detection is delayed when listeners are actively focused on the scenes

rather than listening passively. This latter finding suggests (somewhat counter intuitively) that active listening can counteract the influence of regularity but is consistent with attention acting to reduce 'surprise' (*Spratling, 2008*; *Chennu et al., 2013*).

## Results

### Behavioral data

Listeners' source appearance detection performance in the Active group is shown in *Figure 1B*. Listeners were more accurate and quicker to detect source apperance when the scene structure was temporally regular (REG) versus random (RAND; d' $F_{(1,12)}$ = 100.7, p<0.001; detection times $F_{(1,12)}$ = 17.61, p<0.01). This effect occurred independently of the temporal structure of the appearing component (d' $F_{(1,12)}$ = 0.075, p=0.789; detection times $F_{(1,12)}$ = 4.23, p=0.062). Additionally, listeners were quicker (by ~27 ms) to detect source appearance when it was temporally regular (detection times $F_{(1,12)}$ = 5.70, p<0.050), although this effect did not extend to d' ($F_{(1,12)}$ = 2.29, p=0.156). Thus temporally regular scenes are associated with enhanced detection performance and in a manner independent of the temporal structure of the appearing source. Overall, the mean hit rate was high (mean = 76.1%, ranging from 57 to 97% across listeners) and mean false alarm rate low (mean = 6.25%, ranging from 0 to 18.8%).

### MEG data

#### Scene-evoked response

The neural response evoked by scene onset (i.e. prior to any scene change) is characterized by a series of deflections at around 80, 110 and 200 ms, with topographies corresponding to the commonly observed M50, M100 and M200 onset-response components (*Eggermont and Ponton, 2002*). From around 300 ms post onset, the response settles to a sustained amplitude.

We searched for differences between responses to the onset of REG and RAND scenes using cluster-based permutation statistics (shown in *Figure 2*). Scene temporal structure had a significant effect on the evoked response from 436 ms in the Passive group and from 476 ms in the Active group, involving an increase in the sustained response for REG versus RAND conditions (temporal clusters with FWE corrected significance are indicated as thick horizontal green bars in *Figure 2*; uncorrected clusters are shown as thin light-green bars). The topographical patterns for REG and RAND conditions (averaged over the 500–800 ms period of the sustained response) were qualitatively similar in both Passive and Active groups (also shown in *Figure 2*).

To test whether the onset latency of the scene structure effect was significantly different between groups, we used a jackknife resampling procedure previously shown to be highly sensitive to latency effects (*Miller et al., 1998*; *Ulrich and Miller, 2001*). This involved repeatedly resampling the grand averaged RMS time-course and for each subsample, computing the earliest latency at which the REG versus RAND difference was larger than variability in the baseline period (see Materials and methods). Mean onset latencies for each group are shown as vertical purple lines in *Figure 2* (289 ms for Passive; 412 ms for Active). Although the scene structure effect emerged on average, 123 ms earlier in the Passive versus Active groups, there was no significant difference in onset latency between groups (jackknife adjusted two-sample t(25) = −1.35, p=0.188). Neither was there a main effect of group (p=0.48) or scene structure by group interaction (p=0.31) when conducting ANOVA on the magnitude of the sustained response (averaged from 500–800 ms). This was also the case for earlier time-windows during the M50, M100 and M200 components (all p's >0.19).

In summary, when MEG responses are timelocked to scene onset, regular scene structure results in an increased MEG response from around 400 ms post scene onset. Furthermore, the scene-evoked response shows no evidence of attentional modulation, either in terms of an overall difference between Passive and Active groups or the interaction between scene structure and group.

### Appearance-evoked response

The appearance-evoked response is shown in *Figure 3*. Note that these data have been baseline corrected relative to the 200 ms period prior to the appearance event. Thus, effects reported in this section are specific to the appearance-evoked response and not merely a reflection of the pre-existing REG versus RAND effect observed for the scene-evoked response.

In the Active group, the appearance-evoked response is characterized by a typical pattern of M50/M100/M200 deflections frequently observed at sound onset (as seen above) and following changes within an ongoing sound sequence (*Martin and Boothroyd, 2000*; *Gutschalk et al., 2004*; *Chait et al., 2008*; *Sohoglu and Chait, 2016*). Although the responses here are characterized by later latencies (around 90, 150 and 300 ms, respectively) than those typically observed in other studies that report similar deflections. This may be due to the higher complexity of the present stimuli, which is known to lead to delayed responses (see e.g. *Chait et al., 2008*; *Sohoglu and Chait, 2016*). M50 and M200 deflections are also observed in the Passive group but we note with interest the absence of a prominent M100 component, consistent with previous reports of this component being particularly sensitive to attention and/or task-related demands (*Ahveninen et al., 2011*; *Ding and Simon, 2012*; *Königs and Gutschalk, 2012*; *Sohoglu and Chait, 2016*).

As shown in *Figure 3A*, cluster-based statistics showed a significant effect of scene structure on the appearance-evoked response from 96 ms in the Passive group and from 260 ms in the Active group, both involving an increased neural response for REG versus RAND conditions. This effect was apparent in the Passive group already at the earliest M50 component while in the Active group, it was confined to later components of the evoked response (M200 at corrected significance; M100 uncorrected). As shown in *Figure 3A*, the topographical patterns for REG and RAND conditions were qualitatively similar in both Passive and Active groups.

As described previously, the appearance-evoked response was derived by baseline correcting relative to the 200 ms period prior to source appearance and therefore the measured effect of REG versus RAND is distinct to that observed for the scene-evoked response. To confirm this, we also analyzed matched trials in which there was no change (no appearing source; shown as transparent traces in *Figure 3A*). For this analysis, no effect of REG versus RAND was observed, confirming that scene structure modulates the appearance-evoked response in addition to the scene-evoked response.

The cluster-based permutation statistics above imply an interaction between scene structure and group involving an earlier effect of scene structure in Passive versus Active groups. To directly test this interaction, we estimated the onset latency of the scene structure effect using the jackknife procedure and assessed whether this latency differed significantly between groups. The scene structure effect was estimated to occur on average, 55 ms earlier in Passive versus Active groups (mean onset latency = 87 ms for Passive; 142 ms for Active). This difference was confirmed significant using a jackknife adjusted independent samples test (t(25) = −3.96, p<0.001). This analysis is consistent with the cluster-based permutation statistics above also suggesting a scene structure effect on the early M50 component only in the Passive group. To further characterize this scene structure by group interaction on the M50 peak, a post-hoc between-group t-test (one-tailed) was conducted on the difference in MEG response between REG and RAND conditions at the time of the appearance-evoked M50 (72–112 ms). As shown in *Figure 3B*, the difference in MEG response between REG and RAND conditions was significantly stronger in Passive versus Active groups (t(25) = 1.78, p<0.05.).

An alternative explanation of the interaction between group and scene structure is possible if the M50 and M100 peaks reflect independent but temporally and spatially overlapping components. By this account, the increased response for REG versus RAND scenes at the M50 does not differ between Passive and Active groups. Rather, REG scenes result in an increased M100 in Active listeners that causes a reduction in the M50 (due to their opposite polarities; see topographic plots in *Figure 3A*). To explore this possibility, we selected the twenty most positive and twenty most negative channels at the time of the M50 deflection (72–112 ms; pooling over REG and RAND conditions). The MEG signal was then averaged across channels within these two (positive and negative) groupings and the resulting time-courses analyzed (shown in *Figure 3—figure supplement 1*). As this analysis is based on the mean (rather than RMS) neural response across channels, the polarity of the signal is preserved and thus potentially provides a more accurate representation of the underlying dynamics. Furthermore, the selection of channels based on the M50 deflection would be expected to attenuate interfering responses from the M100 component. As shown in *Figure 3—figure supplement 1*, consistent with the earlier RMS analysis, the M50 showed a larger response for REG versus RAND scenes in Passive but not in Active listeners. This is despite the M100 showing no evidence of modulation by scene structure in Active listeners (even at an uncorrected threshold of p<0.05), making it unlikely the pattern of results reflect a suppression of the M50 by the M100.

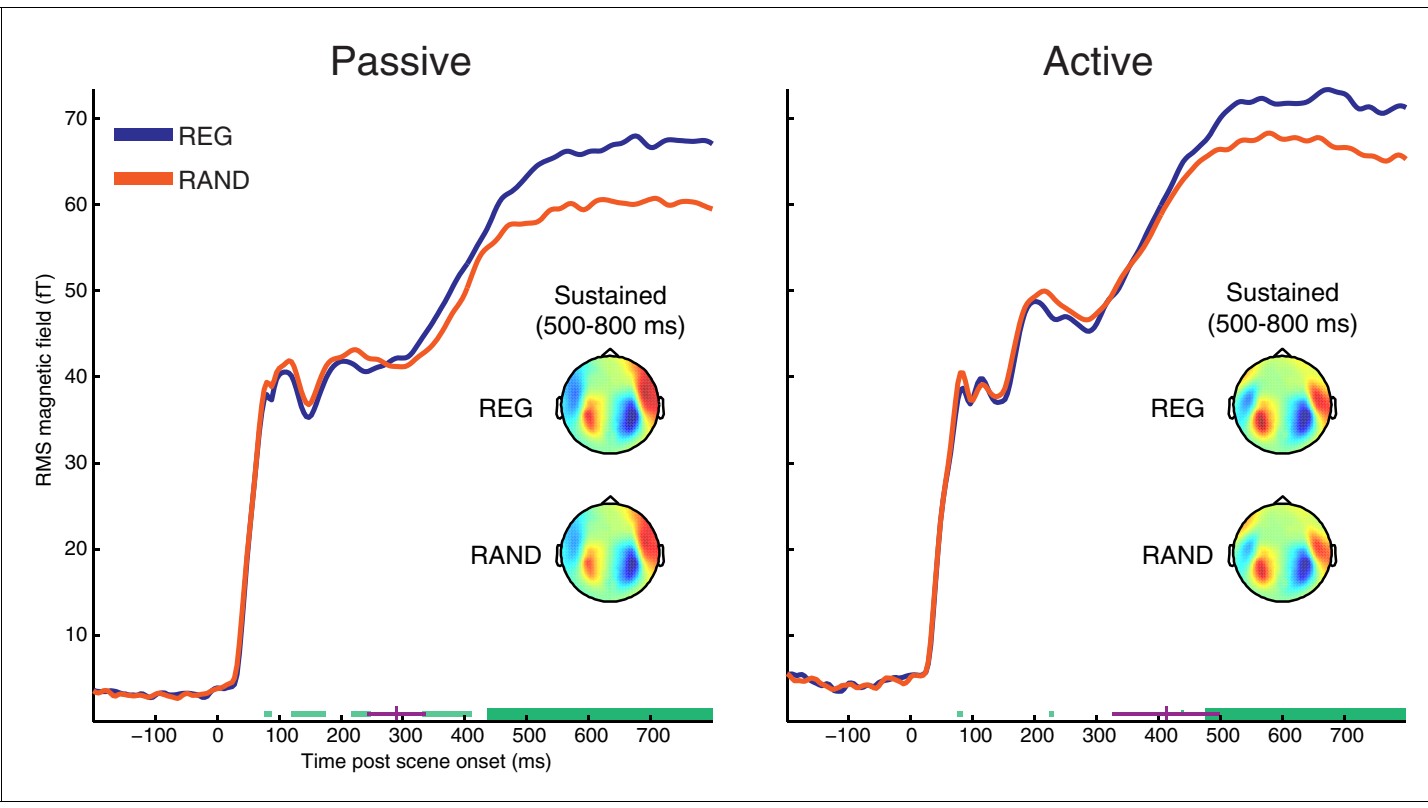

**Figure 2.** RMS time-course of the scene-evoked response showing the main effect of scene temporal structure (REG versus RAND). Thick horizontal green lines indicate time points for which there were significant differences between REG and RAND conditions (p<0.05 FWE corrected at the cluster level; Thin light-green lines show uncorrected clusters). Purple lines indicate (jackknife-estimated) latencies of the onset of the REG versus RAND effect (horizontal and vertical portions indicate mean and jackknife-corrected standard error, respectively). Also shown are topographical patterns at the time of the sustained response (500–800 ms post scene onset), which are characterized by a dipole-like pattern over the temporal region in each hemisphere indicating downward flowing current in auditory cortex (red = source; blue = sink).

The appearance-evoked response as a function of the temporal structure of the appearing source was also analyzed and is shown in *Figure 3C*. Despite listeners' detection times being somewhat quicker when the appearing source was temporally regular versus random (by ~27 ms on average across the group; shown earlier in *Figure 1B*), no significant differences in MEG response were observed in Passive or Active groups. Neither was there a significant interaction between scene and appearing source structure. However, we cannot rule out modulation of more temporally variable neural processes not captured by the evoked analysis employed here. Since the behavioral effects were only observed in detection times, it is also possible that the relevant brain activity is masked by motor response-related processes.

## Source reconstruction

Finally, we localized the neural generators of the scene structure effect. As shown in *Figure 4A*, the scene-evoked response (averaged from 500 to 800 ms) showed greater source power for REG versus RAND scenes in both hemispheres of the superior temporal lobe, including primary auditory cortex, planum temporale and the superior temporal gyrus (peak voxel locations are reported in *Table 1*). An additional distinct cluster of activation is observed in left post central gyrus of the superior parietal lobe.

For the appearance-evoked response, we focused on the scene structure by group interaction ([REG>RAND] > [Passive>Active]) that emerged during the early M50 component. As shown in *Figure 4B*, this effect localized to similar regions as for the scene-evoked response: superior/middle temporal lobe (albeit in the right hemisphere only) and post central gyrus. Additional activation is observed more anteriorly in the pre central gyrus, extending into the middle frontal gyrus.

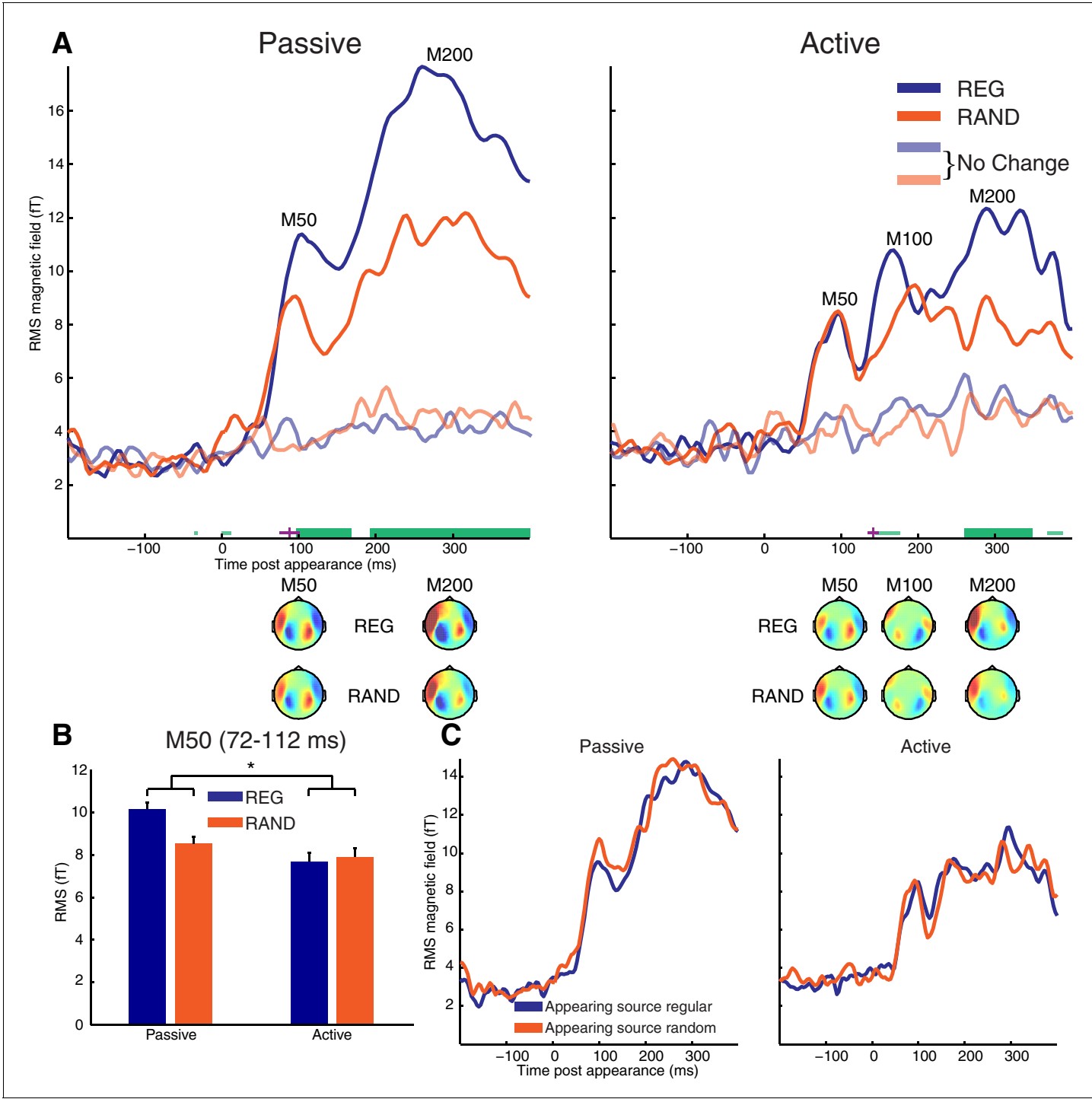

**Figure 3.** Appearance-evoked response. (**A**) RMS time-course of the appearance-evoked response showing the main effect of scene temporal structure (REG versus RAND). Thick horizontal green lines indicate time points for which there were significant differences in RMS between REG and RAND conditions (p<0.05 FWE corrected at the cluster level; Thin light-green lines show uncorrected clusters). Purple lines indicate (jackknife-estimated) latencies of the onset of the REG versus RAND effect (horizontal and vertical portions indicate mean and jackknife-corrected standard error, respectively). Also shown are topographical patterns at the time of the appearance-evoked M50 (72–112 ms), M100 (144–188 ms) and M200 (232–360 ms) components. (**B**) Mean RMS over the appearance-evoked M50 period (712–112 ms). Asterisk indicates the significant (p<0.05) interaction ([REG>RAND]>[Passive>Active]). Error bars represent within-subject standard error of the mean (computed separately for Passive and Active groups. (**C**) Same as panel A but showing main effect of appearing source structure (temporally regular versus random). See also *Figure 3—figure supplement 1* for the MEG time-course averaged over selected sensors responsive to the appearance-evoked M50 component.

*Figure 3 continued on next page*

*Figure 3 continued*

The following figure supplement is available for figure 3:

**Figure supplement 1.** MEG time-course averaged over selected sensors responsive to the appearance-evoked M50 component.

## Discussion

The present study used psychophysics and MEG recordings of brain activity to understand how regular temporal structure facilitates auditory scene analysis. We demonstrate that listeners' ability to detect the appearance of a new source was enhanced in temporally regular scenes. These behavioral benefits of statistical structure on scene analysis are associated with increased neural responses occurring before as well as after source appearance.

### Adaptation versus precision

Around 400 ms following scene onset, we observed an increase in the sustained MEG response for scenes consisting of regularly structured, relative to randomly fluctuating sources. This finding is opposite to what would be expected based on adaptation i.e. *decreased* neural responses for temporally regular events, which has previously been observed for isolated tone sequences (*Costa-Faidella et al., 2011*; *Schwartze et al., 2013*; *Tavano et al., 2014*). It is however consistent with a mechanism that infers the precision (predictability) of sensory input and uses this information to up-regulate neural processing towards more reliable sensory signals (*Feldman and Friston, 2010*; *Zhao et al., 2013*; *Auksztulewicz and Friston, 2015*; *Barascud et al., 2016*). Indeed, it has recently been demonstrated that the magnitude of sustained MEG activity (from naïve distracted listeners) tracks the predictability of rapid tone sequences (*Barascud et al., 2016*). In that study, regularity was characterized by a spectral pattern repeating over time within a single ongoing tone sequence. Although distinct to the temporal regularity studied here, the ensuing effect on MEG response is strikingly similar to the sustained effect we observe. Importantly, the current findings demonstrate mechanisms that automatically (irrespective of directed attention) and rapidly (within 400 ms of scene onset) encode regularities distributed over many concurrent sources, typical of natural listening environments.

If the auditory system can form precise models about the content of ongoing scenes, novel events that violate those models would evoke greater neural responses and be perceived as more salient. Indeed, listeners were better and faster at detecting an appearing source in regular versus random scenes. The MEG response in naïve, passively listening subjects revealed a large (22%) increase in the evoked response starting from the very first response deflection (M50 component) following source appearance. Importantly, this effect occurred over and above that observed prior to the appearance event, demonstrating bottom-up driven 'surprise' responses tightly linked to the predictability of the ongoing scene context. Interestingly, the REG>RAND effect emerged substantially later when participants were actively attending to the appearance events. More discussion of that is below.

Overall, the results demonstrate that the enhanced detection performance observed in behavior is not solely the result of changes in neural responses occurring prior to source appearance (cf. adaptation accounts; *May et al., 1999*; *Jääskeläinen et al., 2004*) but also due to enhanced neural responses to novel events themselves. This is again what would be expected based on precision accounts and is also consistent with animal physiology work showing that the magnitude of responses in single neurons of auditory cortex to new ('deviant') tones is larger than expected based on simple adaptation to previously repeated ('standard') tones alone (*Khouri and Nelken, 2015*).

### Neural sources

Source reconstruction suggests that neural responses in a network of brain regions are modulated by scene temporal structure, including early auditory regions in the superior temporal lobe but also left parietal cortex (post central gyrus). This is consistent with evidence from neuroimaging (*Rao et al., 2001*; *Coull and Nobre, 2008*; *Andreou et al., 2015*), electrophysiology (*Leon et al., 2003*; *Janssen and Shadlen, 2005*) and lesion studies (*Harrington et al., 1998*; *Battelli et al., 2008*) implicating a specific role for left parietal cortex in temporal processing. Parietal cortex has

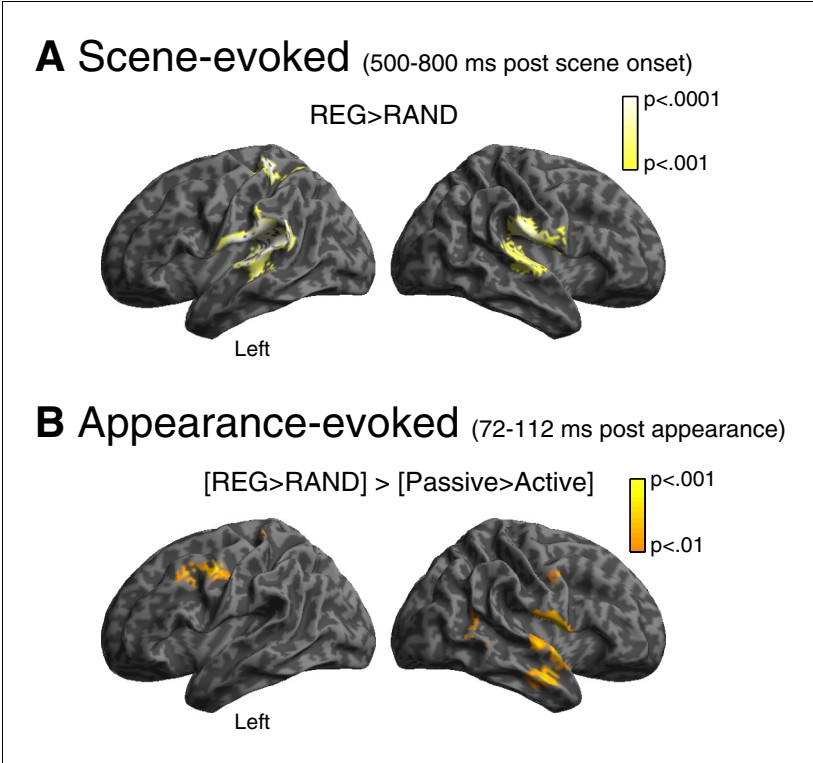

**Figure 4.** Source reconstruction. (**A**) Main effect of scene temporal structure at the time of the sustained portion of the scene-evoked response (500–800 ms post scene onset). Statistical map is overlaid onto an MNI space template brain, viewed over the left and right hemispheres. Color-bar indicates statistical threshold. (**B**) [REG>RAND]> [Passive>Active] interaction at the time of the appearance-evoked M50 component (72–112 ms post appearance).

also been associated with figure-ground processing when the figure is defined by temporally repeatable spectral components in an otherwise randomly structured background (*Teki et al., 2011*; *Teki et al., 2016*). Thus, together the current study and previous findings suggest parietal cortex may be part of a wider network (along with auditory cortical regions) that codes the temporal structure of acoustic scenes. Alternatively, the parietal activity changes we observe may reflect a more domain-general increase in bottom-up saliency attributable to regularity (*Corbetta and Shulman, 2002*; *Zhao et al., 2013*).

We note however that although *Barascud et al. (2016)* report effects of statistical structure in early auditory regions (like the current findings), they did not observe changes in MEG and fMRI responses in parietal cortex. Instead, spectral regularity modulated activity in the inferior frontal gyrus. This is may suggest a degree of neural specialization for the particular type of regularity encoded e.g. temporal-based involving parietal cortex versus spectral-based involving inferior frontal regions. Future work is required however to determine whether temporal and spectral regularities are encoded by distinct neural substrates (e.g. by contrasting neural effects of temporal and spectral regularities in the same experiment).

### Role of attention

Following scene onset (prior to new source appearance), the neural influence of regularity showed no evidence of attentional modulation (the strength of the scene-evoked response to regularly and randomly structured scenes was statistically indistinguishable in passive compared with active listening subjects). This suggests that the brain automatically encodes scene regularities, irrespective of directed attention. After the appearance of a new source, however, regularity and attention had an interactive influence on the evoked response; whereas the first cortical deflection of the appearance-

**Table 1.** Peak voxel locations (in MNI space) and summary statistics from source reconstruction. Activations for the scene-evoked analysis are for the REG>RAND contrast (500–800 ms post scene onset) while those for the appearance-evoked analysis are for the [REG>RAND]>[Passive>Active] interaction contrast (72–112 ms post appearance). Activations have been thresholded using the same parameters as for *Figure 4* (p<0.001 for scene-evoked; p<0.01 for appearance-evoked) but with an additional cluster extent threshold of n > 15 voxels (for display purposes).

| Analysis | Region | Side | Extent | t-value | MNI Coordinates | | |
|---|---|---|---|---|---|---|---|
| | | | | | x | y | z |
| Scene-evoked | Planum Temporale/Parietal Operculum | Left | 1418 | 5.2779 | −48 | −28 | 16 |
| (500-800 ms post scene onset) | | | | 4.364 | −62 | −50 | 14 |
| | | | | 3.9777 | −52 | −30 | -4 |
| | Postcentral Gyrus | Left | 204 | 4.8082 | −32 | −36 | 64 |
| | Supramarginal Gyrus | Right | 704 | 4.2469 | 64 | −24 | 24 |
| | | | | 3.7176 | 44 | −6 | 16 |
| | Planum Temporale | Right | 582 | 3.9459 | 64 | −16 | 6 |
| | | | | 3.9252 | 46 | −26 | 6 |
| | Precentral Gyrus | Right | 19 | 3.6051 | 60 | 6 | 18 |
| Appearance-evoked | Precentral Gyrus | Left | 190 | 3.2219 | −50 | −6 | 44 |
| (72-112 ms post appearance) | | | | 2.9902 | −34 | 6 | 38 |
| | Precentral Gyrus/Central Operculum | Right | 711 | 3.1966 | 56 | 0 | 10 |
| | | | | 3.0153 | 56 | 4 | −10 |
| | | | | 2.9101 | 36 | −8 | 16 |
| | Middle Temporal Gyrus | Right | 157 | 2.9859 | 58 | −2 | −24 |
| | Middle Temporal Gyrus | Right | 55 | 2.6951 | 52 | −54 | 8 |
| | Precentral Gyrus | Right | 21 | 2.6444 | 54 | −4 | 40 |
| | Postcentral Gyrus | Left | 16 | 2.5982 | -30 | −34 | 68 |

evoked response (M50) increased in regular scenes during passive listening, this effect was confined to later deflections (M100 and M200) when listeners actively detected source appearance.

How the neural influence of regularity might depend on attention is the subject of ongoing debate (*Jones and Boltz, 1989*; *Näätänen et al., 2001*; *Summerfield and Egner, 2009*, *2016*; *Winkler et al., 2009*; *Feldman and Friston, 2010*; *Kok et al., 2012*; *Bendixen, 2014*; *Summerfield and de Lange, 2014*; *Schröger et al., 2015*). One proposal is that attention (like regularity) acts to determine the inferred precision of sensory input (*Friston, 2009*; *Barascud et al., 2016*). In this view, attention increases precision (and neural responses) when sensory signals are task-relevant. In this case inferred precision is changed not by the intrinsic structure of the stimulus (e.g. whether temporally regular or random) but by the behavioral goals of the listener. As precision is hypothesized to have a multiplicative (gain) influence on neural activity, this account would have predicted attentional enhancement of the regularity effect. Indeed, *Hsu et al. (2014)* demonstrated greater EEG responses for predictable (ascending) pitch patterns, which was most apparent when those patterns were embedded in an attended stream. In contrast to this pattern, attention in our study delayed the influence of regularity on appearance-related responses.

How then might the current attentional effect be explained? We suggest that attention in our study acted as a form of expectation. That is, when listeners actively detected source appearance, scene changes were relatively more expected. If change-related responses reflect the amount of 'surprise' given the preceding stimulus context, then they should diminish when change is expected and counteract the precision-mediated increase from regularity. Thus, although counterintuitive, the later benefit from regularity when listeners are actively seeking source appearance is consistent with attention acting to reduce surprise.

In the auditory modality, the mismatch negativity (MMN) response is often interpreted as reflecting 'surprise' (*Näätänen et al., 2007*; *Garrido et al., 2009*). The M50 effect we observe occurs earlier and with a distinct topography to the MMN, but may relate to novelty effects on the so-called 'middle-latency' responses (~40 ms) revealed in other work (*Chait et al., 2007*, *2008*; *Grimm et al., 2011*; *Recasens et al., 2014*).

While the proposal that attention acts to reduce surprise may appear at odds with the widespread view of attention playing a distinct functional role to expectation (*Summerfield and Egner, 2009*, *2016*), one associated with enhanced neural processing of attended signals (*Desimone and Duncan, 1995*; *Fritz et al., 2003*), it is consistent with previous observations. In *Chennu et al. (2013)*, listeners were presented with tone sequences containing regularities unfolding over multiple (local and global) timescales. When listeners were instructed to detect deviant tones on a local timescale, the mismatch negativity component indexing that local regularity was attenuated compared with when listeners detected global deviants. The authors interpreted this suppression effect as reflecting reduced surprise from 'top-down expectation (or bias) and consequent attentional focus'. Similarly, *Spratling (2008)* argues that attention and expectation are part of the same general class of top-down signal that act in a similar fashion to modulate perceptual processing. Thus, in this view, the distinction between attention and expectation is blurred and whether these phenomena result in increased or decreased neural processes will depend on the precise details of the stimuli and behavioral demands (*Schröger et al., 2015*; *Henson, 2016*). In this respect, we note that previous investigations of regularity and attention employed static or relatively slow-evolving stimuli (1–5 Hz) and often containing a single perceptual object (image of a face or tone sequence). This may have enabled conscious awareness of stimulus content, involving distinct processes to those relevant to the rapidly evolving and complex scenes employed here and, arguably, to the perceptual challenges faced in natural environments.

## Materials and methods

### Participants

Two groups of participants were tested after being informed of the study's procedure, which was approved by the research ethics committee of University College London. The two groups differed in whether participants' attention was directed away ('Passive' group) or towards ('Active' group) auditory stimulation (see Procedure section below). The Passive group comprised 14 (6 female) participants aged between 19 and 34 years (mean = 23.6, SD = 4.68). All but one of these participants was right-handed. The Active group comprised 13 (7 female), different, right-handed participants aged between 18 and 33 years (mean = 24.3, SD = 4.91). All reported normal hearing, normal or corrected-to-normal vision, and had no history of neurological disorders. There were no significant differences between groups in terms of gender (two-tailed $\chi(1)$ =. 326, p=0.568) or age (two-tailed t (25) = 0.35, p= 0.73).

### Stimuli

Stimuli were 2500–3500 ms duration artificial acoustic 'scenes' populated by seven to eight streams of pure-tones designed to model auditory sources (shown in *Figure 1A*). Each of these sources had a unique carrier frequency (drawn from a pool of fixed values spaced at 2*ERB between 200 and 2800 Hz; *Moore and Glasberg, 1983*) and temporal structure (see below). Previous experiments have demonstrated that these scenes are perceived as composite 'sound-scapes' in which individual sources can be perceptually segregated and selectively attended to, and are therefore good models for listening in natural acoustic scenes (*Cervantes Constantino et al., 2012*). The large spectral separation between neighboring sources (at least two ERBs) was chosen to minimize energetic masking at the peripheral stages of auditory processing (*Moore, 1987*). Signals were synthesized with a sampling rate of 44,100 Hz and shaped with a 30 ms raised cosine onset and offset ramp. They were delivered diotically to the subjects' ears with tubephones (EARTONE 3A 10 Ω, Etymotic Research, Inc) and adjusted to a comfortable listening level.

As shown in *Figure 1A*, a scene change involving the appearance of a new source, could occur partway through the stimulus. The timing of source appearance varied randomly (uniformly distributed between 1000 ms and 2000 ms post scene onset). To facilitate evoked response analysis, the

interval between the time of source appearance and scene offset was fixed at 1500 ms. In the other half of scenes presented, no change occurred ('No Change'). The specific configuration of carrier frequencies and temporal modulation patterns varied randomly across scenes. To enable a controlled comparison between conditions, scenes with and without appearing sources were derived from the same configurations of carrier frequencies and modulation patterns, and then presented in random order during the experiment.

The duration of the tone-pips comprising each source (varying uniformly between 22 and 167 ms) and the silent interval between tone-pips (varying uniformly between 1 and 167 ms) were chosen independently. In 'Regular' (REG) scenes, these tone/silence intervals were fixed so that the temporal structure was regular. This pattern mimics the regularly modulated temporal properties of many natural sounds. In 'Random' (RAND) scenes, tone duration was also fixed but the silent intervals between successive tones varied randomly (with the same distribution as REG scenes i.e. 1–167 ms) resulting in an irregular pattern. Importantly, the above manipulation of scene temporal structure was applied independently of the regularity of the appearing source: Appearing sources could be regular or random (equal proportion), independently of the regularity of the rest of the scene (REG or RAND; equal proportion). Stimuli were randomly ordered during each of eight presentation blocks of 96 trials. The inter-stimulus interval varied randomly between 900 and 1100 ms.

## Procedure

Stimulus delivery was controlled with Cogent software (http://www.vislab.ucl.ac.uk/cogent.php). In the Passive group, participants were naïve to the sounds and engaged in an incidental visual task while looking at a central fixation cross. Participants in this group were instructed to press a button (with their right hand) each time a brief (100 ms duration) image of a pre-defined (target) object appeared on the display at fixation. The target was different on each block and was presented rarely (20%) amongst a stream of non-target images. The inter-image interval ranged from around 500 to 4000 ms and was randomly timed with respect to auditory stimulation. Hit rates ranged from 81 to 95% with false alarm rates below 1%, confirming engagement with the task. In the Active group, participants were instructed to listen carefully to the sounds while looking at a central fixation cross and press a button (with their right hand) as soon as they detected a change in each acoustic scene. Before the experiment, participants in both groups completed a brief (~2.5 min) practice session to familiarize themselves with the task.

## Behavioral statistical analysis

d' scores were obtained for the Active group by first computing for each subject and condition, the hit rate (proportion of source appearances correctly detected) and false alarm rate (proportion of 'No Change' trials for which responses were made). Following this, each d' score was computed as the difference in the z-transformed hit rate and false alarm rate. Detection time was measured between the time of new source appearance and the subject's key press.

## MEG data acquisition and pre-processing

Magnetic fields were recorded with a CTF-275 MEG system, with 274 functioning axial gradiometers arranged in a helmet shaped array. Electrical coils were attached to three anatomical fiducial points (nasion and left and right pre-auricular), in order to continuously monitor the position of each participant's head with respect to the MEG sensors.

The MEG data were analyzed in SPM12 (Wellcome Trust Centre for Neuroimaging, London, UK) and FieldTrip (Donders Institute for Brain, Cognition and Behaviour, Radboud University Nijmegen, the Netherlands) software implemented in Matlab. The data were downsampled to 250 Hz, low-pass filtered at 30 Hz and epoched −200 to 800 ms relative to scene onset (to obtain the scene-evoked response) or −200 to 400 ms relative to the time of the appearance event (to obtain the appearance-evoked response). This epoch encompassed detection-related brain processes leading up to the initiation of the behavioral response in the Active group, which ranged from 465 to 911 ms across participants and conditions. After epoching, the data were baseline-corrected relative to the 200 ms period prior to scene onset (for the scene-evoked data) or prior to the time of source appearance (for the appearance-evoked data).

Subsequent preprocessing differed depending on whether the analysis was conducted in sensor- or source-space. For sensor-space analysis, any trials in which the data deviated by more than three standard deviations from the mean were discarded. Following outlier removal, Denoising Source Separation (DSS) was applied to maximize reproducibility of the evoked response across trials (*de Cheveigné and Simon, 2008*; *de Cheveigné and Parra, 2014*). For each subject, the first two DSS components (i.e., the two 'most reproducible' components; determined −200 to 800 ms relative to scene onset) were retained and used to project both the scene-evoked and appearance-evoked data back into sensor-space, which were then averaged across trials. For source-space analysis, DSS was not performed. Instead, the data were robust averaged across trials to downweight outlying samples (*Wager et al., 2005*; *Litvak et al., 2011*). To remove any high-frequency components that were introduced to the data by the robust averaging procedure, low-pass filtering was repeated after averaging.

Note that although images were presented only in the Passive group, auditory and visual events were temporally uncorrelated. Thus, in both Passive and Active groups, our MEG measures are expected to reflect primarily auditory (and not visual) evoked activity.

## MEG statistical analysis

MEG data across the sensor array were summarized as the root mean square (RMS) across sensors for each time sample within the epoch period, reflecting the instantaneous magnitude of neuronal responses. Group-level paired t-tests were performed for each time sample while controlling the family-wise error (FWE) rate using a non-parametric (cluster-based) permutation procedure based on 5000 iterations (*Maris and Oostenveld, 2007*). Reported effects were obtained by using a cluster defining height threshold of $p<0.05$ with a cluster size threshold of $p<0.05$ (FWE corrected), unless otherwise stated.

Statistical tests of evoked response latency differences were conducted on subsamples of the grand averaged RMS time-course using the jackknife procedure (*Efron, 1981*). In the jackknife procedure, the grand averaged data are resampled *n* times (with *n* being the number of participants) while omitting one participant from each subsample. Statistical reliability of an effect can then be assessed using standard tests (e.g. t-test), not across individual participants, but across subsamples of the grand average. This technique has been shown to be superior to computing latency differences from individual participant data because of the higher signal-to-noise ratio associated with grand averages (*Miller et al., 1998*; *Ulrich and Miller, 2001*). Jackknife-estimated latencies of the scene structure effect were determined by first computing the difference waveform between REG and RAND scenes and then for each jackknife subsample, computing the first latency at which the magnitude of the difference waveform deviated by more than three standard deviations from the mean RMS across time in the baseline period (−200 to 0 ms). When using the jackknife procedure, t-statistics were corrected following the procedure in *Miller et al. (1998)* (multiplication of the subsample standard error by a factor of *n*-1).

To determine the underlying brain sources of the sensor-space effects, we used a distributed method of source reconstruction, implemented within the parametric empirical Bayes framework of SPM12 (*Phillips et al., 2005*; *Litvak and Friston, 2008*; *Henson et al., 2011*). Participant-specific forward models were computed using a Single Shell model and sensor positions projected onto an MNI space template brain by minimizing the sum of squared differences between the digitized fiducials and the MNI template. For inversion of the forward model, we used the 'LOR' routine in SPM12, which assumes that all sources are activated with equal apriori probability and with weak correlation to neighboring sources. This was applied to the entire epoch (−200 to 800 ms for scene-evoked data; −200 to 400 ms for appearance-evoked data).

Source solutions were constrained to be consistent across subjects (pooled over Passive and Active groups), which has been shown to improve group-level statistical power (*Litvak and Friston, 2008*; *Henson et al., 2011*). In brief, this procedure involves 1) realigning and concatenating sensor-level data across subjects 2) estimating a single source solution for all subjects 3) using the resulting group solution as a Bayesian prior on individual subject inversions. Thus, this method exploits the availability of repeated measurements (from different subjects) to constrain source reconstruction. Importantly, however, this procedure does not bias activation differences between conditions in a given source.

Significant effects from sensor-space were localized within the brain (in MNI space, constrained to gray matter) after summarizing source power in the 0–30 Hz range for each participant and time-window of interest using a Morlet wavelet projector (*Friston et al., 2006*). Given that the goal of source reconstruction was to localize the neural generators of sensor-space effects previously identified as significant, statistical maps of source activity are displayed with uncorrected voxelwise thresholds (*Gross et al., 2012*).

## Acknowledgements

This research was supported by a BBSRC project grant BB/K003399/1 awarded to MC. We are grateful to Letty Manyande for technical support during data collection and to Alain de Cheveigné for advice with Denoising Source Separation.

## Additional information

### Funding

| Funder | Grant reference number | Author |
| --- | --- | --- |
| Biotechnology and Biological Sciences Research Council | BB/K003399/1 | Maria Chait<br>Ediz Sohoglu |

The funders had no role in study design, data collection and interpretation, or the decision to submit the work for publication.

### Author contributions

ES, Conception and design, Acquisition of data, Analysis and interpretation of data, Drafting or revising the article; MC, Conception and design, Analysis and interpretation of data, Drafting or revising the article

### Author ORCIDs

Ediz Sohoglu, http://orcid.org/0000-0002-0755-6445
Maria Chait, http://orcid.org/0000-0002-7808-3593

### Ethics

Human subjects: Experimental procedures were approved by the research ethics committee of University College London, and written informed consent was obtained from each participant. Subjects were paid for their participation.

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
