## [Decision Letter]

Thank you for submitting your article "Detecting and representing predictable structure during auditory scene analysis" for consideration by *eLife*. Your article has been reviewed by two peer reviewers – Inyong Choi (Reviewer #1) and Alexander Gutschalk (Reviewer #2) – and the evaluation has been overseen by Andrew King as the Senior and Reviewing Editor.

The reviewers have discussed the reviews with one another and the Reviewing Editor has drafted this decision to help you prepare a revised submission.

Summary:

Using a paradigm designed to simulate the complexity of natural listening environments, this study shows that listeners are more accurate and faster in their ability to detect a new sound source among multiple concurrent sources when the acoustic scene changes in a temporally regular and therefore predictable way than if the changes occur randomly. This behavioral advantage was accompanied by larger and more sustained neural responses, as assessed using magnetoencephalography recordings of brain activity. The change-evoked response was initially weaker in actively attending than passively listening subjects.

Although the reviewers agree that this is high quality and interesting work, they have raised several concerns that will need to be addressed before a final decision can be made.

Essential revisions:

1) It is stated in the third paragraph of the section “MEG statistical analysis” that "source solutions were constrained to be consistent across participants." Please provide more detail about how this was done.

2) The paradigm is sometimes discussed as "detection of a new stream" and sometimes as "change detection". The reviewers recommend that you stick with one description and suggest that this might be referred to as the detection of a new stream in an auditory background.

3) The results suggest that the change/stream onset response is weaker under attention, with the exception of later deflections. The discussion for potential reasons (reduction of surprise under attention) is plausible and interesting. However, could it not be that the amplitude reduction of the positive response is simply caused by the appearance of the negative response and the cancellation between the two (cf. Figure 3 in Sohoglu and Chait, 2016)? This could be explored e.g. by calculating difference waves between attended and not-attended conditions.

4) Parietal activity is discussed here in the context of regularity processing. However, in Barascud, Chait et al. 2016 – which is also related to regularity detection – frontal, but no parietal activity was observed. You should probably be more cautious in the interpretation of these different patterns of activity in a distributed source model.

---

## [Author Response]

*1) It is stated in the third paragraph of the section “MEG statistical analysis” that "source solutions were constrained to be consistent across participants." Please provide more detail about how this was done.*

We now provide additional information on this procedure. Group-optimization of subject-specific source inversions is a widely used feature of SPM’s Bayesian source reconstruction tools (for full details of the method, see Litvak and Friston 2008 NeuroImage; Henson et al. 2011 Front Hum Neurosci. For example applications, see Furl et al. 2011 NeuroImage; Spitzer et al. 2011 PNAS; Sohoglu et al. 2012 J Neurosci).

We now provide more details of this procedure in the revised manuscript (fourth paragraph of the section “MEG statistical analysis”):

“Source solutions were constrained to be consistent across subjects (pooled over Passive and Active groups), which has been shown to improve group-level statistical power (Litvak and Friston, 2008; Henson et al., 2011). In brief, this procedure involves 1) realigning and concatenating sensor-level data across subjects 2) estimating a single source solution for all subjects 3) using the resulting group solution as a Bayesian prior on individual subject inversions. Thus, this method exploits the availability of repeated measurements (from different subjects) to constrain source reconstruction. Importantly, however, this procedure does not bias activation differences between conditions in a given source.”

*2) The paradigm is sometimes discussed as "detection of a new stream" and sometimes as "change detection". The reviewers recommend that you stick with one description and suggest that this might be referred to as the detection of a new stream in an auditory background.*

We agree that using a single description would enhance clarity. We have followed the reviewers’ suggestion and now use ‘detection of appearing source’ throughout the revised manuscript.

3) The results suggest that the change/stream onset response is weaker under attention, with the exception of later deflections. The discussion for potential reasons (reduction of surprise under attention) is plausible and interesting. However, could it not be that the amplitude reduction of the positive response is simply caused by the appearance of the negative response and the cancellation between the two (cf. Figure 3 in Sohoglu and Chait, 2016)? This could be explored e.g. by calculating difference waves between attended and not-attended conditions.

We thank the reviewers for this excellent point. In the revised manuscript, we include additional analysis (Results section, subsection “Appearance-evoked response”) and a new supplementary figure (Figure 3—figure supplement 1) to address this issue:

“An alternative explanation of the interaction between group and scene structure is possible if the M50 and M100 peaks reflect independent but temporally and spatially overlapping components. […] This is despite the M100 showing no evidence of modulation by scene structure in Active listeners (even at an uncorrected threshold of p<0.05), making it unlikely the pattern of results reflect a suppression of the M50 by the M100.”

*4) Parietal activity is discussed here in the context of regularity processing. However, in Barascud, Chait et al. 2016 – which is also related to regularity detection – frontal, but no parietal activity was observed. You should probably be more cautious in the interpretation of these different patterns of activity in a distributed source model.*

We agree that the parietal activity should be interpreted within the limits of MEG source localization accuracy and that to conclusively determine whether there are genuine differences in the neural substrates between studies, the two types of regularity (temporal-based in current study versus spectral-based in Barascud, Chait et al. 2016) would need to be statistically contrasted in a single experiment.

This is now acknowledged in the revised manuscript (Discussion section, subsection “Neural sources”):

“We note however that although Barascud et al. (2016) report effects of statistical structure in early auditory regions (like the current findings), they did not observe changes in MEG and fMRI responses in parietal cortex. Instead, spectral regularity modulated activity in the inferior frontal gyrus. This is may suggest a degree of neural specialization for the particular type of regularity encoded e.g. temporal-based involving parietal cortex versus spectral-based involving inferior frontal regions. Future work is required however to determine whether temporal and spectral regularities are encoded by distinct neural substrates (e.g. by contrasting neural effects of temporal and spectral regularities in the same experiment).”